# Dual Coupled Long-Range Hybrid Surface Plasmon Polariton Waveguide for Sub-Wavelength Confinement

**DOI:** 10.3390/mi14122167

**Published:** 2023-11-29

**Authors:** Yindi Wang, Shulong Wang, Juanning Zhao, Mingyuan Xue

**Affiliations:** 1School of Electronic Information and Artificial Intelligence, Shaanxi University of Science and Technology, Xi’an 710021, China; zxychangchun@126.com (J.Z.); xuemy@sust.edu.cn (M.X.); 2Key Laboratory for Wide Band Gap Semiconductor Materials and Devices of Education, School of Microelectronics, Xidian University, Xi’an 710071, China; slwang@xidian.edu.cn

**Keywords:** hybrid waveguide, SPPs, mode coupling, optoelectronic integrated systems

## Abstract

In this paper, a long-range hybrid waveguide for subwavelength confinement based on double SPP coupling is proposed. The hybrid waveguide consists of a metal-based cylindrical hybrid waveguide and a silver nanowire. There are two coupling regions in the waveguide structure that enhance mode coupling. Strong mode coupling enables the waveguide to exhibit both a small effective mode area (0.01) and an extremely long transmission length (700 μm). The figure of merit (FOM) of the waveguide can be as high as 4000. In addition, the cross-sectional area of the waveguide is only 500 nm × 500 nm, allowing optical operation in the subwavelength range, which helps enhance the miniaturization of optoelectronic devices. The excellent characteristics of the hybrid waveguide make it have potential applications in photoelectric integrated systems.

## 1. Introduction

With the rapid development of optoelectronic integrated systems, the performance of optical devices is required to be higher [1,2]. A smaller device size is needed to improve circuit integration [3,4,5]. As the core component of an optoelectronic integrated system, waveguides need not only a longer transmission distance but also an extremely small mode area to enhance the integration and miniaturization of photoelectric integrated circuits [6,7,8,9]. In this context, designing sub-wavelength waveguides with excellent performance becomes a great challenge [10,11,12].

Traditional waveguides have a long transmission distance but poor light localization, which is not conducive to the design of small-sized devices [13,14]. Surface plasmon polaritons (SPPs) are a special electromagnetic wave excited at the interface between metal and medium [15,16,17,18]. SPP waves have excellent localization and can limit light to the range of sub-wavelength [19,20,21,22,23], which can be used to design small-sized and even sub-wavelength devices [24,25,26,27]. The combination of traditional waveguides with SPPs can be used to design waveguides with a long transmission distance and a small effective mode area. Nowadays, there are many kinds of SPP hybrid waveguides, and their performance has been greatly improved [28,29,30,31,32,33,34,35]. But with the rapid development of optoelectronic integrated systems, the requirement for waveguides is increasing [36,37,38]. New breakthroughs must be found to achieve good performance on small-sized devices.

In this paper, a high-performance hybrid waveguide at 1550 nm was designed using the double-reinforced coupling of metal SPP mode and silicon waveguide mode. The hybrid waveguide consists of a metal-based cylindrical hybrid waveguide and a silver nanowire. The two coupling regions in the waveguide structure enhance the modal coupling. The strong mode coupling makes the light field locally compact, and the waveguide shows an extremely small effective mode area. Therefore, the hybrid waveguide combines the long transmission length of the silicon waveguide with the small mode area of the SPP wave. In addition, the waveguide realizes the sub-wavelength operation of light.

## 2. Structure Design

The hybrid waveguide consists of a metal-based cylindrical hybrid waveguide and silver nanowires. Figure 1a is a schematic diagram of a metal-based cylindrical hybrid waveguide. The cylindrical silver nanowire passes through the center of the silicon waveguide, and the void region is filled with silicon dioxide. The radius of the silver nanowire is labeled *r*_Ag_. The thickness of the silicon dioxide and silicon layer is labeled as *t*_SiO2_ and *t*_Si_, respectively.

Figure 1b shows the hybrid waveguide proposed in this paper. A silver nanowire with a radius of *R*_Ag_ was placed above the metal-based cylindrical hybrid waveguide. The spacing between silver nanowire and metal-based cylindrical hybrid waveguides is *g*. The whole structure of the hybrid waveguide is covered by SiO_2_ cladding. In this paper, the waveguide was studied at the 1550 nm communication wavelength.

## 3. Methods

The proposed metal-based cylindrical hybrid waveguide realizes the double coupling of silicon waveguide mode and SPP mode in its structure. The superposition and strengthening of coupling are helpful to improve the waveguide characteristics, making the performance of the waveguide better than other existing hybrid waveguides.

A waveguide is an important part of an optoelectronic system, and its performance affects other waveguide-based optoelectronic devices [39,40]. In practical applications, the requirements of a waveguide include a small size, a long transmission distance, and a small mode area. The main parameters used to evaluate waveguide characteristics include effective mode area, transmission length, figure of merit, and modal effective index [41,42]. The formulas for calculating each parameter are discussed in detail below.

Effective mode area (*A_eff_*) characterizes the localization of light in a waveguide. The value of *A_eff_* is expected to be as small as possible in the design of the waveguide. A smaller *A_eff_* means that the light field is more concentrated in the waveguide and quickly disappears in the direction perpendicular to the transmission (longitudinal). *A_eff_* is calculated by the following formula [14,42]:(1)$$A_{eff}=\iint_{-\infty }^{+\infty }W\left(x,y\right)dxdy/max\left(W\left(x,y\right)\right)$$

According to the above formula, *A_eff_* is the ratio of total energy to maximum energy density. The electromagnetic energy density can be calculated using the following formula [43]:(2)$$W\left(x,y\right)=\frac{1}{2}Re\left\{\frac{d\left[\omega \varepsilon \left(x,y\right)\right]}{d\omega }\right\}{\left|E\left(x,y\right)\right|}^{2}+\frac{1}{2}\mu_{0}{\left|H\left(x,y\right)\right|}^{2}$$
where *H*(*x*,*y*) is the magnetic field and *E*(*x*,*y*) is the electric field. The normalized effective modal area *A_eff_*/*A*_0_, where *A*_0_ = *λ*^2^/4, is commonly used to evaluate the characteristics of waveguides.

The transmission length (*L_m_*) of a waveguide is defined as the distance that light travels when the energy decays to 1/e of its initial value. *L_m_* can be calculated using the following equation [8,44]:(3)Lm=λ/4πImNeff
where *N_eff_* is the modal effective index, *N_eff_* = *n_eff_* + I × *α_eff_*. αeff is the imaginary part of *N_eff_*, which represents the loss of light in the waveguide. It can be defined as *α_eff_* = Im(*N_eff_*). *A_eff_* and *L_m_* are two opposing parameters. A smaller effective modal area results in a shorter transmission length. There is generally an appropriate balance between *L_m_* and *A_eff_*. Thus, the figure of merit (*FOM*) provides a measure of compromise. The calculation formula for *FOM* is as follows:(4)$$FOM=\frac{L_{m}}{2\sqrt{\frac{A_{eff}}{\pi }}}$$

## 4. Results and Discussion

In this paper, the metal-based cylindrical hybrid waveguide was examined first, which consists of a cylindrical silver nanowire passing through the center of the silicon waveguide with a SiO_2_ gap between the silver nanowire and Si. The good characteristics of the waveguide were obtained through mode coupling of the silicon photonic mode with the silver nanowire SPP mode.

The waveguide was studied at the working wavelength of 1550 nm. We improved the waveguide’s performance by optimizing its geometric parameters. The characteristic parameters and electric field distribution of the waveguide were simulated by COMSOL Multiphysics software 5.4. At the 1550 nm working wavelength, the relative permittivity of Ag, Si, and SiO_2_ is −129 + 3.3i, 12.25, and 2.25, respectively [42,45].

Firstly, the propagation characteristics of waveguides with different silver nanowire radii (*r*_Ag_) were studied. In this simulation, *t*_SiO2_ and *t*_Si_ are fixed at 5 nm and 25 nm, and *r*_Ag_ varies from 5 nm to 50 nm. As illustrated in Figure 2a,b, the normalized effective mode area (*A_eff_*/*A*_0_) is very high below 20 nm, decreases rapidly after 20 nm, and then increases slightly after 25 nm. *L_m_*, however, has the opposite trend. The main reason for these phenomena is that when *r*_Ag_ is less than 20 nm, the radius of silver nanowire is too small and no mode coupling occurs, so the mode area is large and the optical transmission length is extremely short because of the heavy loss. After 25 nm, the coupling weakens with the increase in *r*_Ag_, so *A_eff_/A*_0_ increases.

Figure 2c–j shows the electric field distributions of waveguides at different silver nanowire radii (*r*_Ag_), and the electric field distribution refers to the magnitude of the total electric field. It can be seen that when *r*_Ag_ = 20 nm, the electric field distribution is diffused in the cladding, and no mode coupling occurs. When *r*_Ag_ is equal to or greater than 25 nm, mode coupling occurs, and a strong electric field is concentrated in the SiO_2_ gap region. With the increase in *r*_Ag_, mode coupling weakens, and electric field localization weakens. These results agree well with the calculated data in Figure 2a,b. In the subsequent simulation, *r*_Ag_ is set to 25 nm.

Next, the characteristics of the waveguide on SiO_2_ thicknesses (*t*_SiO2_) were studied. In this simulation, *r*_Ag_ is fixed at 25 nm and *t*_Si_ is fixed at 30 nm, and *t*_SiO2_ varies from 1 nm to 20 nm. The results are shown in Figure 3a,b. It can be seen that with the increase in SiO_2_ thickness, the normalized effective mode area (*A_eff_*/*A*_0_) and transmission length of the hybrid waveguide increase slightly. This is because larger SiO_2_ gaps will result in weaker mode coupling.

Figure 3c–f shows the electric field distribution of the metal-based cylindrical hybrid waveguide at different *t*_SiO2_. It can be seen that the electric field in Figure 3c is well localized, and electric field localization gradually weakens with the increase in *t*_SiO2_, which is in good agreement with Figure 3a,b. According to these simulation results, the optimal size of *t*_SiO2_ is 5 nm.

Then, the influence of silicon thickness (*t*_Si_) on waveguide characteristics was studied. *r*_Ag_ and *t*_SiO2_ are fixed at 25 nm and 5 nm, respectively. *t*_Si_ ranges from 5 nm to 50 nm. The results are shown in Figure 4. When *t*_Si_ is less than 35 nm, both the normalized effective mode area and the transmission distance decrease gradually, which is because the increase in silicon thickness leads to the enhancement of mode coupling, then optical localization increases and transmission length decreases. When *t*_Si_ is greater than 35 nm, the normalized effective modal area increases rapidly, and the transmission distance decreases rapidly. This is because there is no mode coupling in the waveguide when *t*_Si_ is in this interval, and light field dispersion causes serious loss, so the transmission distance is extremely small.

Figure 4c–j shows the electric field distributions at different silicon thicknesses (*t*_Si_). As illustrated in Figure 4c–i, with the increase in *t*_Si_, mode coupling becomes stronger, and the electric field becomes more localized. Figure 4j shows the electric field distribution at *t*_Si_ = 40 nm. It can be seen that no mode coupling occurs at *t*_Si_ = 40 nm, and the electric field is dispersed throughout the waveguide structure. These results are consistent with Figure 4a,b. The optimal value of *t*_Si_ is 35 nm.

Then, on the basis of the metal-based cylindrical hybrid waveguide, we introduced dual-mode coupling, investigated its performance in detail, and optimized its geometric parameters. The structure of the dual-coupled hybrid waveguide is shown in Figure 1b. According to the simulation results of the metal-based cylindrical hybrid waveguide, the optimized structural parameters are *t*_Si_ = 35 nm, *r*_Ag_ = 25 nm, *t*_SiO2_ = 5 nm, and *R*_Ag_ = 30 nm. Firstly, the influence of gap height (*g*) on hybrid waveguide performance was studied. As shown in Figure 5, with the increase in gap height (*g*), both the normalized effective modal area and the transmission length increase. Figure 5c shows the quality factor of the waveguide (*FOM*). Figure 5d shows the loss of light in the waveguide (*α_eff_*), and its change trend is opposite that of *L_m_*. The main reason is that the mode coupling weakens with the increase in g, resulting in the weakening of optical localization, so *A_eff_*/*A*_0_ increases and the transmission length of light increases. It can be seen that the double coupling of modes improves the performance of the waveguide significantly. The transmission length increases by an order of magnitude while the normalized mode area remains small, and *FOM* increases by 20 times.

Figure 6 shows the electric field distributions of waveguides at different gap heights (*g*). We can see that with an increase in *g*, the localization of the electric field decreases, and the electric field distribution becomes more dispersed. Due to the double coupling of the modes, the electric field is mainly concentrated in the SiO_2_ gap region between silver and silicon. The electric field is most concentrated at *g* = 5 nm. The results are consistent with Figure 5, so the best value for *g* is 5 nm.

Then, the influence of silver nanowire radius (*R*_Ag_) on waveguide performance was studied. As shown in Figure 7, it can be seen that with the increase in *R*_Ag_, the effective mode area and transmission length both decrease due to the enhancement of mode coupling. It can be concluded that an ultra-small area (0.01) and an ultra-long transmission length (700 μm) can be obtained by optimizing *R*_Ag_, and *FOM* can reach 4000.

To further verify the above calculation results, Figure 8 lists the electric field distributions of the waveguide at different silver nanowire radii. It can be seen that as *R*_Ag_ increases, the electric field becomes more concentrated, and mode coupling becomes stronger. According to the results in Figure 7 and Figure 8, we made a compromise between *A_eff_*/*A*_0_ and *L_m_* and finally chose the value of *R*_Ag_ as 30 nm.

Based on the above research results, it can be concluded that the optimal size of the proposed dual-coupled hybrid waveguide is *r*_Ag_ = 25 nm, *t*_SiO2_ = 5 nm, *t*_Si_ = 35 nm, *g* = 5 nm, and *R*_Ag_ = 30 nm. Under these conditions, the hybrid waveguide shows excellent waveguide characteristics.

## 5. Conclusions

In conclusion, we proposed a hybrid waveguide based on metal SPPs at 1550 nm. By adopting two coupling regions in the waveguide structure, extremely strong mode coupling is achieved. The results show that the hybrid waveguide has excellent performance, such as a small effective mode area of 0.01 and a long transmission length of 700 μm. In addition, the figure of merit (*FOM*) can be as high as 4000, and the cross-sectional area of the waveguide is 500 nm × 500 nm, which enables optical manipulation in the sub-wavelength range. These excellent characteristics enable our proposal to have great potential in optoelectronic integrated systems.

## Figures and Tables

**Figure 1 micromachines-14-02167-f001:**
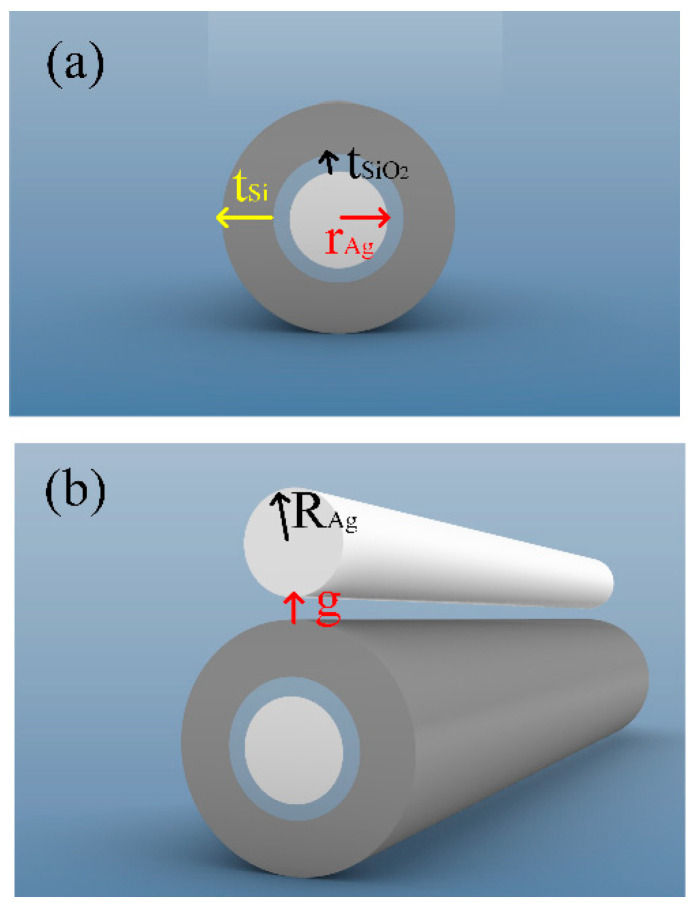
The stereograph view of the proposed structure: (**a**) a metal-based cylindrical hybrid waveguide and (**b**) the proposed hybrid waveguide.

**Figure 2 micromachines-14-02167-f002:**
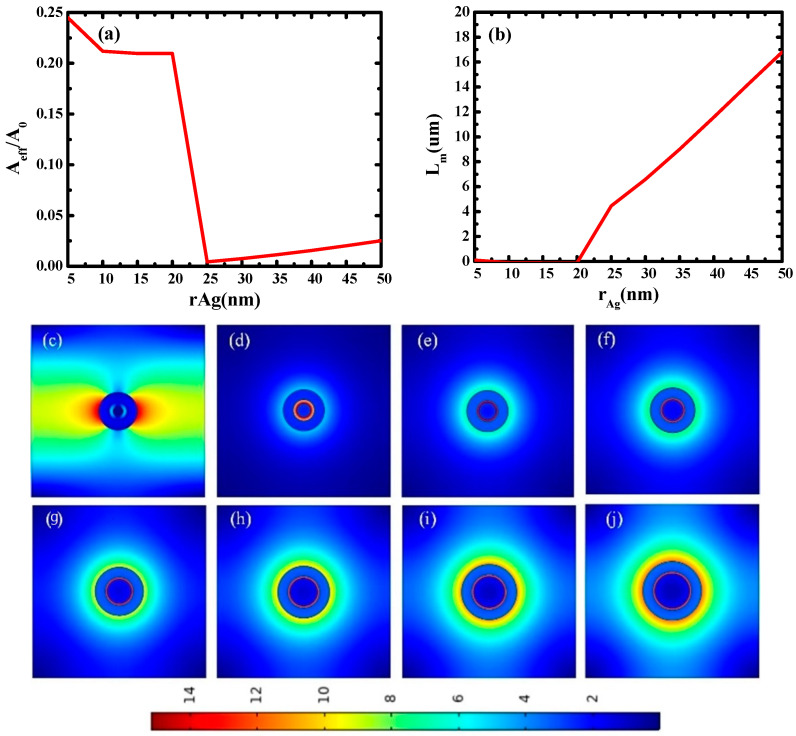
Dependence of waveguide characteristics on silver nanowire radius (*r*_Ag_): (**a**) normalized mode area (*A_eff_*/*A*_0_) and (**b**) propagation length (*L_m_*); electric field distributions of waveguides at different *r*_Ag_: (**c**) *r*_Ag_ = 20 nm; (**d**) *r*_Ag_ = 25 nm; (**e**) *r*_Ag_ = 30 nm; (**f**) *r*_Ag_ = 35 nm; (**g**) *r*_Ag_ = 40 nm; (**h**) *r*_Ag_ = 45 nm; (**i**) *r*_Ag_ = 50 nm; (**j**) *r*_Ag_ = 55 nm.

**Figure 3 micromachines-14-02167-f003:**
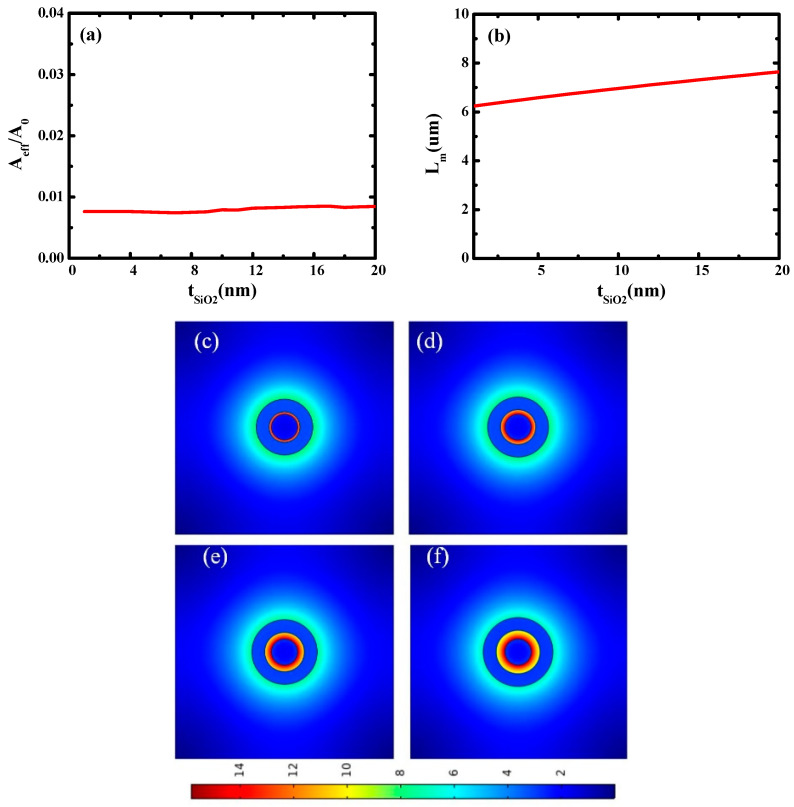
(**a**) Dependence of waveguide characteristics on SiO_2_ thicknesses (*t*_SiO2_): (**a**) normalized mode area (*A_eff_/A*_0_) and (**b**) propagation length (*L_m_*); electric field distributions at different SiO_2_ thicknesses (*t*_SiO2_): (**c**) *t*_SiO2_ = 5 nm; (**d**) *t*_SiO2_ = 10 nm; (**e**) *t*_SiO2_ = 15 nm; (**f**) *t*_SiO2_ = 20 nm.

**Figure 4 micromachines-14-02167-f004:**
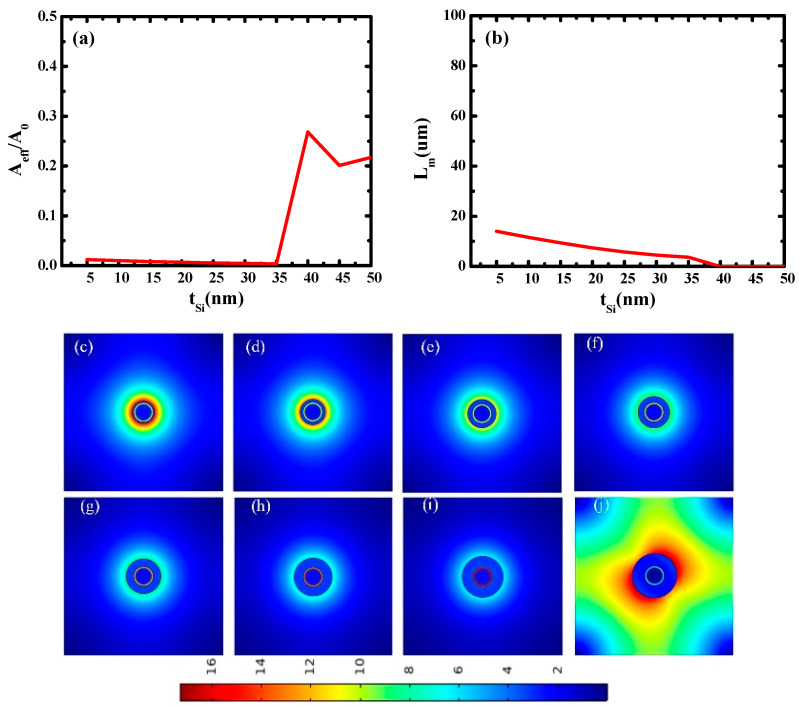
Dependence of waveguide characteristics on silicon thickness (*t*_Si_): (**a**) normalized mode area (*A_eff_/A*_0_) and (**b**) propagation length (*L_m_*); electric field distributions at different *t*_Si_: (**c**) *t*_Si_ = 5 nm; (**d**) *t*_Si_ = 10 nm; (**e**) *t*_Si_ = 15 nm; (**f**) *t*_Si_ = 20 nm; (**g**) *t*_Si_ = 25 nm; (**h**) *t*_Si_ = 30 nm; (**i**) *t*_Si_ = 35 nm; (**j**) *t*_Si_ = 40 nm.

**Figure 5 micromachines-14-02167-f005:**
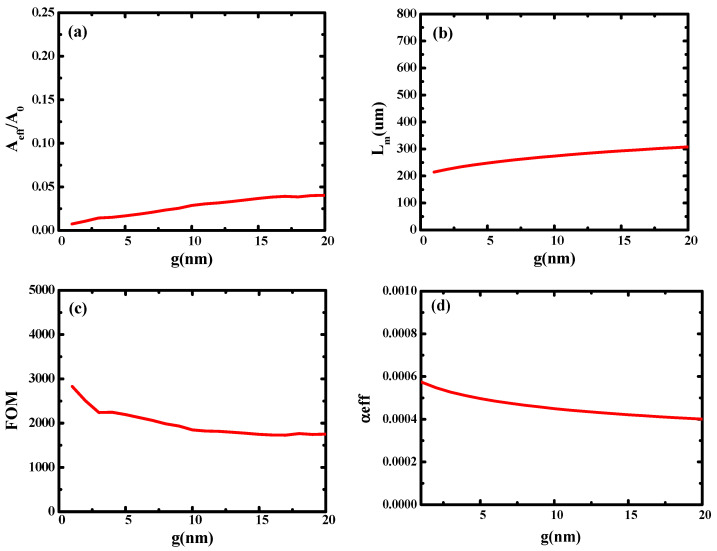
Dependence of waveguide characteristics on gap height (*g*): (**a**) normalized mode area (*A_eff_*/*A*_0_); (**b**) propagation length (*L_m_*); (**c**) the quality factor of the waveguide (*FOM*); (**d**) the loss of light in the waveguide (*α_eff_*).

**Figure 6 micromachines-14-02167-f006:**
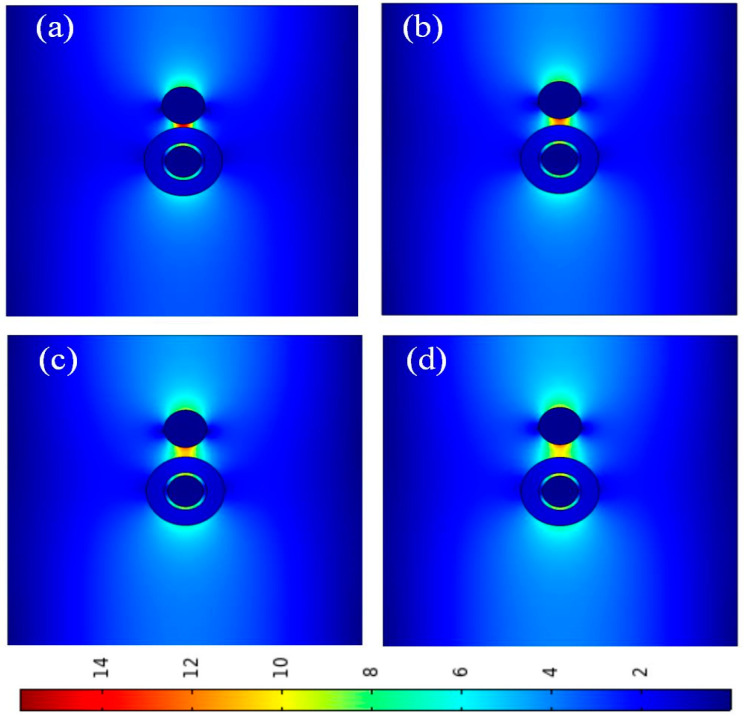
Electric field distributions at different gap heights (*g*): (**a**) *g* = 5 nm; (**b**) *g* = 10 nm; (**c**) *g* = 15 nm; (**d**) *g* = 20 nm.

**Figure 7 micromachines-14-02167-f007:**
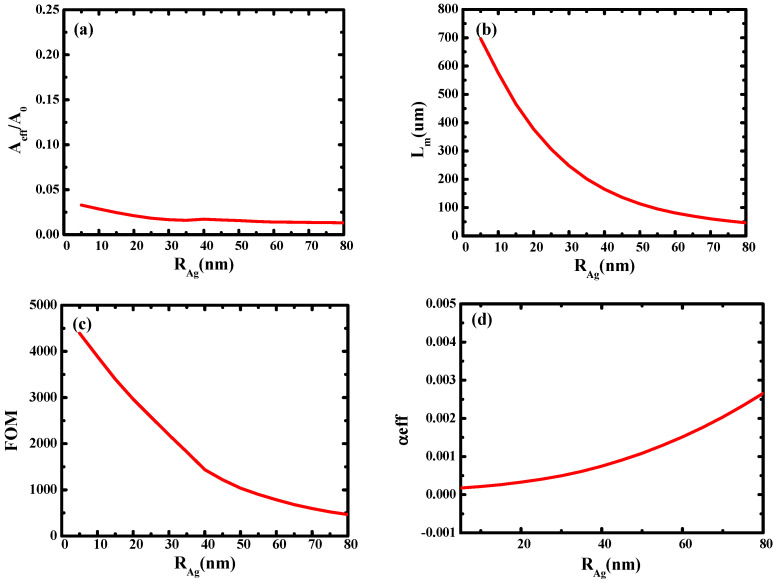
Dependence of waveguide characteristics on the silver nanowire radius (*R*_Ag_): (**a**) normalized mode area (*A_eff_*/*A*_0_); (**b**) propagation length (*L_m_*); (**c**) the quality factor of the waveguide (*FOM*); (**d**) the loss of light in the waveguide (*α_eff_*).

**Figure 8 micromachines-14-02167-f008:**
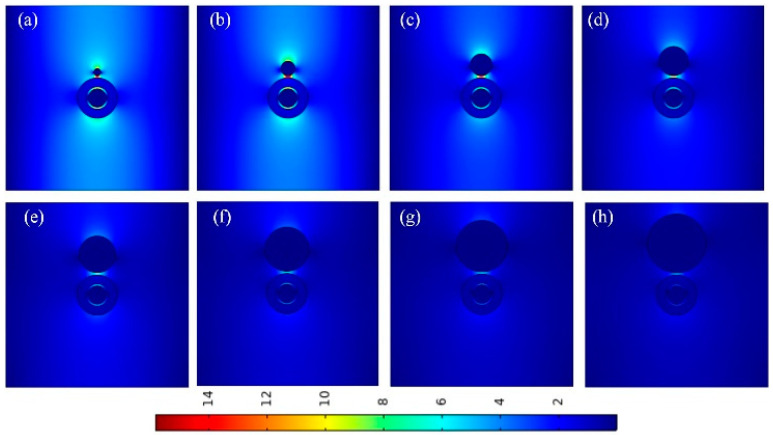
Electric field distributions at different *R*_Ag_: (**a**) *R*_Ag_ = 10 nm; (**b**) *R*_Ag_ = 20 nm; (**c**) *R*_Ag_ = 30 nm; (**d**) *R*_Ag_ = 40 nm; (**e**) *R*_Ag_ = 50 nm; (**f**) *R*_Ag_ = 60 nm; (**g**) *R*_Ag_ = 70 nm; (**h**) *R*_Ag_ = 80 nm.

## Data Availability

Data are contained within the article.

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
