# Peer review of "Dual Coupled Long-Range Hybrid Surface Plasmon Polariton Waveguide for Sub-Wavelength Confinement"

_micromachines, 2023, doi:10.3390/mi14122167_

Round 1

Reviewer 1 Report

Comments and Suggestions for Authors

Yindi et al. report a long range hybrid waveguide consisting of a metal-based cylindrical hybrid waveguide and a silver nanowire, where long transmission distance of 700 µm and effective mode area of 0.01 as well as FoM of 4000 were reported. I recommend publishing this manuscript only after revising based on the following concerns:

(1)    In the abstract: “Subwavelength Confinement” should be subwavelength confinement.

(2)    Eq.1. The definition of A in dA should be specified. The using of A as a variable is not advised since A is the modal area.

(3)    Eq. 3, In my opinion, the definition of Lm is not described appropriately. Lm= ? ⁄[2 ? Im(neff) is the distance that light travels when the electric field decays to 1/e of the initial value, see Journal
of Materials Chemistry C, 2020, 8(20): 6832-6838.
Lm= ? ⁄[4 ? Im(neff) is the distance that light travels when the energy decays to 1/e of the initial value, see IEEE Journal of Selected Topics in Quantum Electronics, 2013, 20(4): 181-188.

(4)    “silicon photon” should be “silicon photonic mode”

(5)    In Figs.2,3,6,8 the color bars should be corrected.

(6)    In Figs.5, 6, What is the value of RAg?

(7)    Below eq.4 and in Fig.5 and 7. The Lm plots of Figs.5b and 7b already contain the Im(Neff). However, the authors made a mistake, such as the neff plots of Figs.5d and 7d. While neff should be the modal effective index, and neff is always larger than 1. We suggest the authors define Neff = neff+i αeff. In this case, αeff = Im(Neff) represents the loss of light in waveguide. neff is the modal effective index.

(8)    Plasmonic waveguides have been widely investigated at different wave bands based on different materials. In the introduction part, the authors missed to refer to work on recent achievements of plasmon waveguides based on graphene, transistion metal dichalcogenides, bulk Dirac semimetals and akali metals, such as :
Journal of Optics 25(6) (2023) 065802;
Nanoscale 11(10) (2019) 4601-4613;
Optics Letters 46(3) (2021) 472-475.
Nano Letters 23(15) (2023) 7150-7156;
Nanomaterials 2022, 12(12), 1950; 
Nature 581(7809) (2020) 401-405.
Optics Communications 499 (2021) 127316. https://doi.org/10.1007/s11468-023-02114-2.

(9)  Grammatical errors should be corrected, such as “The thickness of silicon dioxide and silicon layer are labeled…”; “Silver nanowire with radius RAg are …”; “The spacing between silver nanowires…”, etc.

Comments on the Quality of English Language

none

Author Response

Title: Dual coupled long-range hybrid SPPs waveguide for Sub-wavelength Confinement

Authors: Yindi Wang1,*, Shulong Wang2, Juanning Zhao1 and Mingyuan Xue1

Journal: Micromachines

Manuscript Number: micromachines-2729374

Reviewer: 1

Point #1: In the abstract: “Subwavelength Confinement” should be subwavelength confinement.

Respond:

Thank you very much. We've corrected it.

Point #2: Eq.1. The definition of A in dA should be specified. The using of A as a variable is not advised since A is the modal area.

Respond:

We have changed the expression of Eq.1 as follows:

Point #3: Eq. 3, In my opinion, the definition of Lm is not described appropriately. Lm= ? ⁄[2 ? Im(neff) is the distance that light travels when the electric field decays to 1/e of the initial value, see Journal of Materials Chemistry C, 2020, 8(20): 6832-6838.  Lm= ? ⁄[4 ? Im(neff) is the distance that light travels when the energy decays to 1/e of the initial value, see IEEE Journal of Selected Topics in Quantum Electronics, 2013, 20(4): 181-188.

Respond:

Thank you very much. We have changed the definition of Lm to: the distance that light travels when the energy decays to 1/e of the initial value.

Point #4: “silicon photon” should be “silicon photonic mode”.

Respond:

Thank you. We have corrected this mistake.

Point #5: In Figs.2,3,6,8 the color bars should be corrected.

Respond:

Thank you. The color bars of these electric field distribution diagrams are the result of COMSOL software simulation. There will be slight differences between the color bars of each graph.

Point #6: In Figs.5, 6, What is the value of RAg?

Respond:

Thank you very much. In Figs.5, 6, the initial value of RAg is 30 nm.

Point #7: Below eq.4 and in Fig.5 and 7. The Lm plots of Figs.5b and 7b already contain the Im(Neff). However, the authors made a mistake, such as the neff plots of Figs.5d and 7d. While neff should be the modal effective index, and neff is always larger than 1. We suggest the authors define Neff = neff+i αeff. In this case, αeff = Im(Neff) represents the loss of light in waveguide. neff is the modal effective index.

Respond:

Thank you very much. Neff=neff+i*aeff. aeff is the imaginary part of Neff, which represents the loss of light in waveguide. It can be defined as aeff=Im (Neff). Fig.5 and 7 have been changed in the manuscript.

Point #8: Plasmonic waveguides have been widely investigated at different wave bands based on different materials. In the introduction part, the authors missed to refer to work on recent achievements of plasmon waveguides based on graphene, transistion metal dichalcogenides, bulk Dirac semimetals and akali metals, such as : Journal of Optics 25(6) (2023) 065802; Nanoscale 11(10) (2019) 4601-4613; Optics Letters 46(3) (2021) 472-475. Nano Letters 23(15) (2023) 7150-7156; Nanomaterials 2022, 12(12), 1950; Nature 581(7809) (2020) 401-405. Optics Communications 499 (2021) 127316. https://doi.org/10.1007/s11468-023-02114-2.

Respond:

Thank you very much. We have added these documents to the manuscript.

[31]         Y. Qin et al.Highly confined low-loss light transmission in linear array-enabled hybrid plasmonic waveguides. Journal of Optics. 2023,vol. 25, p. 065802.

[32]         K. Zheng et al.Ultra-high light confinement and ultra-long propagation distance design for integratable optical chips based on plasmonic technology. Nanoscale. 2019,vol. 11, pp. 4601-4613.

[33]         X. He.;F. Liu.;F. Lin.;W. Shi.Tunable 3D Dirac-semimetals supported mid-IR hybrid plasmonic waveguides. Optics Letters. 2021,vol. 46, pp. 472-475.

[34]         Z. Gao et al.Low-Loss Plasmonics with Nanostructured Potassium and Sodium–Potassium Liquid Alloys. Nano letters. 2023,vol. 23, pp. 7150-7156.

[35]         D. Teng et al.Sodium-based cylindrical plasmonic waveguides in the near-infrared. Nanomaterials. 2022,vol. 12, p. 1950.

[26]         Y. Wang et al.Stable, high-performance sodium-based plasmonic devices in the near infrared. Nature. 2020,vol. 581, pp. 401-405.

[27]         J. Gao.;C. Hou.;F. Wang.;H. Liu.;T. Ma.A directional coupler based on graphene-enhanced Na-loaded plasmonic rib waveguide. Optics Communications. 2021,vol. 499, p. 127316.

Point #9: Grammatical errors should be corrected, such as “The thickness of silicon dioxide and silicon layer are labeled…”; “Silver nanowire with radius RAg are …”; “The spacing between silver nanowires…”, etc.

Respond:

Thank you very much. These grammatical errors have be corrected. “The thickness of silicon dioxide and silicon layer are labeled as…”; “Silver nanowire with radius RAg is …”; “The spacing between silver nanowire…”

Reviewer 2 Report

Comments and Suggestions for Authors

In the paper micromachines-2729374 "Dual coupled long-range hybrid SPPs waveguide for Sub-wavelength Confinement" by Yindi Wang and Shulong Wang  authors consider the dual core plasmonic waveguide, consisting of two parallel silver circular waveguides, one of which is covered with SiO2 shell and Si shell. Using numerical simulations authors optimize parameters, in order to minimize the effective mode area and maximize transmission  length. The paper can be published after some minor revisions.

  1. In line 91 neff is introduced, but it is never used. Maybe introduce it after Eq.(3) and rewrite it?
  2. In Fig.4a the behaviour of curve is non-monotonic, making some local maximum. Why it appears?
  3. In the text Figs.5c and 5d are never referred.

Author Response

Title: Dual coupled long-range hybrid SPPs waveguide for Sub-wavelength Confinement

Authors: Yindi Wang1,*, Shulong Wang2, Juanning Zhao1 and Mingyuan Xue1

Journal: Micromachines

Manuscript Number: micromachines-2729374

Reviewer: 2

Point #1: In line 91 neff is introduced, but it is never used. Maybe introduce it after Eq.(3) and rewrite it?

Respond:

Thank you very much. We put the introduction of Neff after Eq.(3). “where Neff is the modal effective index, Neff=neff+i*aeff. aeff is the imaginary part of Neff, which represents the loss of light in waveguide. It can be defined as aeff=Im (Neff)”.

Point #2: In Fig.4a the behaviour of curve is non-monotonic, making some local maximum. Why it appears?

Respond:

Thank you very much. When tSi is greater than 35 nm, there is no mode coupling in waveguide when tSi is in this interval, and the light field dispersion causes serious loss. At this time, the change of effective mode area is different from that when mode coupling occurs, and is not monotonic.

Point #3: In the text Figs.5c and 5d are never referred

Respond:

Fig. 5(c) shows the quality factor of the waveguide (FOM). Fig. 5(d) shows the loss of light in waveguide (aeff), and its change trend is opposite to that of Lm. It can be seen that the double coupling of modes improves the performance of the wave-guide significantly. The transmission length increases by an order of magnitude while the normalized mode area remains small, and FOM increased by 20 times.

Reviewer 3 Report

Comments and Suggestions for Authors

Please see attached pdf document.

Comments on the Quality of English Language

It may be helpful to ask for professional editing service to make the flow of the manuscript better.

Author Response

Title: Dual coupled long-range hybrid SPPs waveguide for Sub-wavelength Confinement

Authors: Yindi Wang1,*, Shulong Wang2, Juanning Zhao1 and Mingyuan Xue1

Journal: Micromachines

Manuscript Number: micromachines-2729374

Reviewer: 3

Point #1: The authors defined the effective mode area using Eq. (1). While this definition has been used to characterize the localization of mode profiles for hybrid waveguides, it is just one of the ways to quantify mode confinement. Other definitions, like Eq. (2) in [I.D. Rukhlenko et al. Opt. Let. 37, 2295-2297 (2012)] can also be used to calculate the effective mode area and may be superior since it avoids the use of maximum energy density, which can exhibit singular behavior for certain plasmonic waveguide geometries. The authors should discuss how the alternative definitions of effective mode area could change the results and conclusions of this study.

Respond:

Thank you very much. There are indeed multiple ways to calculate the effective mode area, and the waveguides presented in this manuscript are conventional geometries, so both the definition in Eq. (1) or the method in [I.D. Rukhlenko et al. Opt. Let. 37, 2295-2297 (2012)] are applicable.

Point #2: The authors introduced the concept of Figure of Merit ( FOM) using Eq. (4), without justifying it. One could also use alternative definitions such as (Lm/lambda)/(Aeff/lambda^2) as the FOM, which can produce different results and affect conclusions. The authors may want to elaborate on why a particular definition of FOM is used and whether other definitions of FOM should be considered.

Respond:

Thank you very much. Both Eq. (4) and (Lm/lambda)/(Aeff/lambda^2)  can be used to characterize the Figure of Merit, and the value will be different, and the meaning of the representation is the same. The calculation formulas Eq. (4) used in this manuscript refer to the following sources.

[1] Dong,Lu,Lei,et al.Hybrid Tube-Triangle Plasmonic Waveguide for Ultradeep Subwavelength Confinement[J].Journal of Lightwave Technology: A Joint IEEE/OSA Publication, 2017, 35(11):2259-2265.

[2] Buckley R, Berini P .Figures of merit for 2D surface plasmon waveguides and application to metal stripes [J].Optics Express, 2007, 15(19):12174.DOI:10.1364/OE.15.012174.

Point #3: On page 8, line 202 and 203, the authors stated “According to the results in Fig. 7 and Fig. 8, we made a compromise between Aeff/A0 and Lm, and finally chose the value of RAg as 30 nm.” But according to the FOM plot in Figure 7(c), the optimal value for RAg should be 5 nm. This shows that the FOM definition adopted by the authors is not a good measure of the overall performance of the waveguide. The authors may want to elaborate on what their “compromise between Aeff/A0 and Lm” means.

Respond:

Thank you very much. The figure of merit (FOM) provides a measure of compromise, we comprehensively consider the value of FOM and the geometric size of the device, because considering the actual manufacturing in the later stage, the size is too small is not conducive to the actual process manufacturing. Therefore, we chose a RAg value of 30 nm to ensure both high FOM value and actual preparation.

Point #4: This study was focused on the properties of the fundamental mode of the hybrid waveguide design. However, in certain cases the higher-order modes may offer beter performance

especially when both mode confinement and propagation length are taken into consideration, such as in [J. Colanduoni et al. Plasmonics 11, 763–769 (2016)]. It may be worth commenting on the properties of higher-order modes.

Respond:

Thank you very much. The SPPs hybrid waveguide proposed in the manuscript has strong mode coupling, only the first order mode, that is, the main mode, is considered. The higher-order modes is not considered because the strength of the higher-order modes is small, and it is quickly lost in the waveguide and the transmission distance is small.

Point #5: In Figures 2, 3, 4, 6, and 8, the authors stated that the color plots are for the electric field distributions. But it is not clearly whether it refers to the magnitude of the total electric field, the square of the magnitude of the total electric field, or just the magnitude of some component. The authors may want to make it clear in the figure caption, and label it next to the color bar.

Respond:

Thank you for your advice. We have added the statement to the manuscript. The electric field distributions in Figures 2, 3, 4, 6, and 8 refer to the total electric field, and the color depth represents the magnitude of the total electric field of each point in the waveguide.

Point #6: The aspect ratios in Figure 3(c)-(f) do not seem to be correct as the waveguide cross-sections are not circular.

Respond:

Thank you. This error has been corrected. The waveguide cross-sections are not circular, which is caused by my manual adjustment of the size of the picture.

Point #7: The range for the y-axis in Figures 5(d) and 7(d) may be reduced so that the variation in neff can become clearly visible.

Respond:

Thank you. You are right. Narrowing down the Y-axis can reveal the obvious change trend of aeff, which is opposite to that of normalized mode area (Aeff/A0) and propagation length (Lm).

Point #8: On page 5, line 142, the authors stated that “tSi ranges from 5 nm to 30 nm.” But the range of tSi is actually 5 nm to 50 nm in Figure 4.

Respond:

Thank you very much. We have changed it “tSi ranges from 5 nm to50 nm.”

Round 2

Reviewer 3 Report

Comments and Suggestions for Authors

The authors have addressed most of my questions. Regarding the use of effective mode area to quantify mode confienemnt, besides the one used by the authors, other definitions, like Eq. (2) in [I.D. Rukhlenko et al. Opt. Lett. 37, 2295-2297 (2012)] can also be used to calculate the effective mode area and may be superior. In the revised manuscript, the authors should consider to briefly discuss why they use the current defnition of effective mode area and how alternative definitions of effective mode area could change the results and conclusions of this study.

Comments on the Quality of English Language

 Minor editing of English language is recommended.

Author Response

Title: Dual coupled long-range hybrid SPPs waveguide for Sub-wavelength Confinement

Authors: Yindi Wang1,*, Shulong Wang2, Juanning Zhao1 and Mingyuan Xue1

Journal: Micromachines

Manuscript Number: micromachines-2729374

Reviewer: 3

Point #1: The authors have addressed most of my questions. Regarding the use of effective mode area to quantify mode confienemnt, besides the one used by the authors, other definitions, like Eq. (2) in [I.D. Rukhlenko et al. Opt. Lett. 37, 2295-2297 (2012)] can also be used to calculate the effective mode area and may be superior. In the revised manuscript, the authors should consider to briefly discuss why they use the current defnition of effective mode area and how alternative definitions of effective mode area could change the results and conclusions of this study.

Respond:

Thank you very much. Both Eq. (1) in the manuscript and Eq. (2) in [I.D. Rukhlenko et al. Opt. Lett. 37, 2295-2297 (2012)] are used to calculate the effective pattern area, but they are applicable to different objects. Eq. (2) in [I.D. Rukhlenko et al. Opt. Lett. 37, 2295-2297 (2012)] is inapplicable to plasmonic waveguides, where time-averaged of energy density has different signs inside metal and dielectric, and thus the total power flow may vanish. Eq. (1) in the manuscript is applicable to plasmonic waveguides, and is derived from References 43 and 44, the definition of which is explained in detail in References 43 and 44. How do other definitions of effective modal area change the results and conclusions of this study, which we will examine in detail later

[43]Dong et al.Hybrid Tube-Triangle Plasmonic Waveguide for Ultradeep Subwavelength Confinement. Hybrid tube-triangle plasmonic waveguide for ultradeep subwavelength confinement. 2017,vol. 35, pp. 2259-2265.

[44]Y. Bian.;Q. Gong.Deep‐subwavelength light confinement and transport in hybrid dielectric‐loaded metal wedges. Laser Photonics Reviews. 2014,vol. 8, pp. 549-561